# Intelligent Rapid Detection Techniques for Low-Content Components in Fruits and Vegetables: A Comprehensive Review

**DOI:** 10.3390/foods13071116

**Published:** 2024-04-06

**Authors:** Sai Xu, Yinghua Guo, Xin Liang, Huazhong Lu

**Affiliations:** 1Institute of Facility Agriculture, Guangdong Academy of Agricultural Sciences, Guangzhou 510640, China; liangxin@gdaas.cn; 2College of Engineering, South China Agricultural University, Guangzhou 510642, China; gyhmqdqxygyqya@163.com; 3Guangdong Academy of Agricultural Sciences, Guangzhou 510640, China

**Keywords:** fruits and vegetables, intelligent rapid detection, low-content components

## Abstract

Fruits and vegetables are an important part of our daily diet and contain low-content components that are crucial for our health. Detecting these components accurately is of paramount significance. However, traditional detection methods face challenges such as complex sample processing, slow detection speed, and the need for highly skilled operators. These limitations fail to meet the growing demand for intelligent and rapid detection of low-content components in fruits and vegetables. In recent years, significant progress has been made in intelligent rapid detection technology, particularly in detecting high-content components in fruits and vegetables. However, the accurate detection of low-content components remains a challenge and has gained considerable attention in current research. This review paper aims to explore and analyze several intelligent rapid detection techniques that have been extensively studied for this purpose. These techniques include near-infrared spectroscopy, Raman spectroscopy, laser-induced breakdown spectroscopy, and terahertz spectroscopy, among others. This paper provides detailed reports and analyses of the application of these methods in detecting low-content components. Furthermore, it offers a prospective exploration of their future development in this field. The goal is to contribute to the enhancement and widespread adoption of technology for detecting low-content components in fruits and vegetables. It is expected that this review will serve as a valuable reference for researchers and practitioners in this area.

## 1. Introduction

With the burgeoning development of the world’s economy, there is a growing concern regarding food quality. Fruits and vegetables, considered daily essentials, are no longer pursued solely for texture and flavor but have garnered attention for their nutritional value. These essential ingredients encompass numerous low-content components, including beneficial elements like vitamins and minerals [1,2], offering timely nutritional supplementation. Concurrently, they also harbor potentially harmful components such as heavy metals, fungi, and pesticide residue [3,4], posing a significant threat to human health if consumed in excess. Timely and precise detection of these low-content components in fruits and vegetables is imperative for safeguarding human health.

Conventional methods for detecting low-content components in fruits and vegetables, such as spectrophotometric, titration, atomic absorption spectrometry (AAS), atomic fluorescence spectrometry (AFS), and inductively coupled plasma-mass spectrometry (ICP-MS), often involve complex preprocessing, cumbersome detection procedures, long analysis times, and demanding requirements for inspectors. These factors hinder the real-time monitoring of fruits and vegetables. However, advancements in science and technology have led to the emergence of new rapid detection methods that simplify the process and reduce analysis times. While intelligent technologies have made notable strides in detecting high-content substances like sugar, moisture, and starch content, there is limited research on the detection of low-content components. Moreover, the accuracy of intelligent rapid detection technology remains a focal point of current research. Despite these advancements, intelligent rapid detection technology still has some drawbacks, including lower accuracy compared to traditional methods and higher costs. Therefore, continuous improvement is necessary to refine and enhance this technology. This paper aims to explore the current development status, applications, and existing challenges of intelligent rapid detection technology while proposing future research directions to inspire further advancements in the field.

## 2. Low-Content Components in Fruits and Vegetables

### 2.1. Nutrients in Fruits and Vegetables

Fruits and vegetables contain various essential components that are not just limited to protein, sugar, and fat. These components, including trace minerals like Fe, Zn, Cu, and Mn, as well as vitamins A, C, and E, amino acids, flavonoids, polyphenols, and other bioactive compounds, play a significant role in maintaining human health [5,6]. Minerals are vital for regulating physiological functions such as substance metabolism, oxygen transportation, and cell signaling, ensuring the proper functioning of organs in the human body. Minerals are also key components that makeup bones and teeth, helping to maintain the strength and stability of the bone structure [7,8,9]. On the other hand, vitamins are life-sustaining components that maintain normal body functions by regulating various biochemical processes and metabolic pathways within cells. Different types of vitamins play different roles in cells, including participation in energy metabolism, cell signaling, DNA synthesis, immune function, maintenance of cellular structure, etc., which are essential for maintaining normal physiological functions of the body [10,11,12]. In addition, people can also absorb many nutrients from fruits and vegetables, such as anthocyanins, carotene, flavonoids, and other low-content components, all of which play a crucial role in maintaining health. Despite their low content in food, these nutrients are indispensable for promoting human health. It is crucial to detect these nutrients in fruits and vegetables in a timely and rapid manner. This not only improves the quality of produce in the market but also safeguards the health of the population.

### 2.2. Ingredients Required for the Growth of Fruits and Vegetables

Low-content components in fruits and vegetables not only play a vital role in human health but also contribute significantly to plant growth and maturation. One example is gibberellin, which can be found in tomatoes and peppers in varying concentrations, ranging from micrograms to milligrams. It stimulates the growth of plant stems and promotes flowering and fruiting [13]. Minerals like magnesium act as activators for various enzymes in plant metabolism and nucleic acid synthesis. Iron, on the other hand, is primarily found in plant chloroplasts and has an indirect impact on plant photosynthesis, ensuring efficient utilization of light energy for energy conversion. Therefore, the timely detection of these low-content components in fruits and vegetables is crucial for monitoring plant growth [14,15]. By closely monitoring these elements, farmers can make necessary adjustments to their planting programs, ensuring plants receive adequate nutrition. This, in turn, enhances fruit yield and quality, contributing to the scientific management of agricultural production.

### 2.3. Toxic Residues in Fruits and Vegetables

Fruits and vegetables often contain toxic components, which are attributed to the use of pesticides in their cultivation. Farmers rely on pesticides to prevent insects, control weeds, and clear pathogens. However, excessive pesticide usage leads to the presence of pesticide residues in these crops. In China, the maximum residue limit (MRL) for organophosphorus pesticides in fruits and vegetables falls between 0.01 mg/kg and 0.5 mg/kg, according to the national standard GB2763-2021 [16]. Consuming fruits and vegetables with pesticide residues poses significant health risks. These pesticide chemicals, if the dose is too large, will lead to acute poisoning in the human body, and in severe cases, it may even be life threatening. Long-term exposure to low doses of pesticides can also result in chronic health problems, such as damage to the nervous system, immune system, and an overburdened liver [17,18,19,20]. Furthermore, pesticide residues absorbed by the liver during plant growth, along with the uptake of heavy metal substances from the soil, can result in heavy metal poisoning, cellular cancer, and chronic diseases. Therefore, the implementation of intelligent and rapid detection technology for harmful substances in fruits and vegetables before they reach the market is crucial for enhancing food safety and preventing health issues caused by these harmful substances. The development of such detection methods is urgently needed [21,22,23,24].

Research and detection of low-content components in fruits and vegetables has received significant attention due to the increasing complexity of food safety concerns. To address the growing demand for accurate and efficient detection methods, there is a need for intelligent rapid detection technology. This technology is expected to provide a more precise and efficient way of detecting low-content components to quickly and accurately detect harmful substances in fruits and vegetables, so as to screen them and ensure the health problems of the residents’ diet. The goal of research in this field is to bridge existing gaps and provide comprehensive support for public health protection.

## 3. Rapid and Intelligent Detection Technology for Low-Content Components

### 3.1. Near-Infrared Spectroscopy (NIR)

NIR is mainly due to the non-resonant nature of molecular vibrations. When near-infrared light illuminates a sample, the molecules absorb light at a specific frequency, which causes the molecules to vibrate or rotate, producing leaps in the molecular energy levels. These jumps result in a reduction in the intensity of the transmitted light. The relevant information is captured and converted into electrical signals through optical fibers, and these signals are then transmitted to a spectrometer to form a near-infrared spectrum. The spectra are analyzed using chemometrics, from which the relevant information on low-content components in fruits and vegetables can be extracted and describe the chemical composition of an unknown mixture or food. Then, the detection model is established by combining spectral data and machine learning. By inputting the spectral data of fruits and vegetables into the detection model, rapid and intelligent detection can be achieved. Reflectance occurs when a sample is illuminated by a light source, absorbing some of the light while reflecting the rest. The receiver then captures the intensity of this reflected light. In transmission mode, the light source passes through the sample, where it is partially absorbed and scattered, with the remaining portion penetrating the sample and reaching the receiver, which records the intensity of the transmitted light. In diffuse reflectance mode, NIR light is directed onto the sample surface, where some of it is reflected. The receiver then captures and records the intensity of this reflected light, forming a reflectance spectrum of the sample. Reflectance is commonly utilized to assess surface characteristics, allowing for the evaluation of fruit and vegetable appearance quality, defects, and ripeness by analyzing the properties of the reflected light. Meanwhile, transmission enables light to penetrate the sample, providing insights into its internal organization and composition. Diffuse reflectance involves light that has been scattered multiple times within the sample, yielding more comprehensive information, used for analyzing the chemical composition, nutritional content, and tissue structure of fruits and vegetables [25,26,27].

NIR is a valuable technique for analyzing chemical bonds, including C–O, O–H, N–H, and S–H, among others, thus providing valuable insights into the composition of organic matter and compounds found in samples. Compared to traditional detection methods, NIR spectroscopy offers several advantages such as no preprocessing requirements, quick detection times, easy operation, and absence of pollution. Its effectiveness has been demonstrated in the detection of various low-content components, including carotenoids, polyphenols, fatty acids, and thioglucosides, in a wide range of fruits and vegetables [28].

### 3.2. Hyperspectral Imaging (HSI)

Hyperspectral imaging (HSI) technology is a rapid, nondestructive, and intelligent detection technique with timely results. The HSI detection device depicted in Figure 1 consists of a hyperspectral camera, an imaging spectrometer, a fiber optic halogen lamp, and a computer [29]. The technology is based on spectral images; HSI combines spectral data and computer images when a halogen lamp emits a beam of light on the sample, and the hyperspectral camera receives the light reflected by the sample, takes pictures, and captures the optical properties of the same crop over a continuous wavelength range.

HSI technology enables the extraction of internal compositional details of fruits and vegetables by capturing two-dimensional spectral information. On the other hand, computer images provide physical information such as external size, shape, and color, facilitating intelligent classification of these agricultural products.

HSI technology offers numerous advantages, particularly in terms of high-resolution and real-time capabilities. Currently, extensive research is being conducted to explore the application of HSI technology in the detection of low-content components in fruits and vegetables. This research focuses on estimating various internal chemical composition parameters such as nitrogen, phosphorus, chlorophyll, leaf area index, yield, and water content. By employing HSI technology for real-time monitoring of crop growth, valuable information regarding the physiological state and internal composition of crops can be obtained [30,31,32,33].

Hyperspectral spectroscopy, compared to near-infrared spectroscopy, offers a greater capacity for capturing a larger number of wavelength bands. This leads to the generation of extensive and high-dimensional spectral data, which provides more detailed information about the internal composition of fruits and vegetables. However, dealing with such vast data introduces certain challenges, including redundant information and interference. To address these challenges and ensure accurate analysis, it becomes necessary to employ various techniques for extracting wavelengths and preprocessing the data. Additionally, the construction of precise prediction models based on this extensive data requires careful consideration and fine-tuning.

### 3.3. Raman Spectroscopy

Raman spectroscopy consists of Rayleigh scattering and Raman scattering. Scattering occurs when a laser beam of a specific frequency strikes the surface of a sample. In most cases, the scattered light has the same frequency as the incident light, and this scattering, known as Rayleigh scattering, is a type of elastic scattering in which no exchange of energy occurs during the process. However, there is another case where some of the scattered light has a different frequency than the incident light, which is inelastic scattering due to interaction with the sample, called Raman scattering. Since Raman spectroscopic detection is based on inelastic scattering, Rayleigh scattering does not provide useful information. Raman scattering detects samples by inelastic scattering of molecular vibrations. Figure 2 illustrates the basic principle of Raman scattering: when a molecule in its ground state is irradiated by a laser, it is excited and polarized. Electrons jump to higher energy levels and then return to lower energy levels, resulting in a change in the frequency of the scattered light. A decrease in frequency is called a Stokes line and an increase in frequency is called an anti-Stokes line. By analyzing Raman spectroscopic data (including vibrational frequencies, peak position changes, and peak widths), important information about the biochemical state and compositional structure of a sample can be obtained [34,35,36].

Raman spectroscopy is known for its ability to detect low-content components in fruits and vegetables, but its main limitation is its weak signal strength. In 1974, Fleischmann et al. discovered that the adsorption of pyridine on a rough silver electrode could greatly enhance the Raman scattering signal. Building on this, Moskovits et al. proposed that the collective oscillation of electrons in nanostructures on metal surfaces contributes to the signal enhancement in Raman spectroscopy. As a result, the technique of surface-enhanced Raman spectroscopy (SERS) emerged, which integrates precious metals to amplify Raman signals [37,38,39,40]. Both Raman spectroscopy and SERS have become powerful tools for detecting low-content components in fruits and vegetables. The application of Raman spectroscopy has expanded to include the detection of heavy metal ions, microorganisms, nutrients, and pesticide residues [41,42].
Figure 2Energy level diagram of Raman scattering [43]. Reprinted with permission from Ref. [43]. 2011, Taylor & Francis.
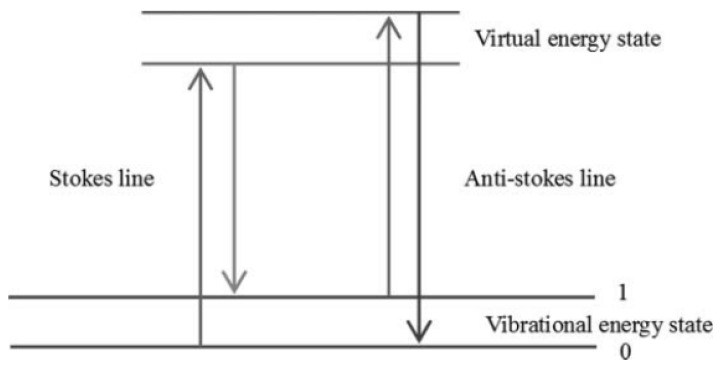



### 3.4. Laser Induced Breakdown Spectroscopy (LIBS)

LIBS is presented in Figure 3. This technique utilizes a high-power pulsed laser to excite a sample, causing it to undergo excitation through numerous pulses in a short period. This process leads to the formation of a weak electron plasma. After the plasma diffuses and cools down, the excited atomic and ionic species within the plasma emit an optical signal. The signal is then collected and transmitted to a spectrometer. By analyzing the spectral information collected by the spectrometer, a computer creates a characteristic spectrum. The composition of the tested samples, along with stoichiometry, can be rapidly determined by analyzing these characteristic spectra. This enables quantitative and qualitative analysis of the target elements [44,45].

The LIBS technique suffers from several limitations such as poor stability, inaccurate measurement, insufficient sensitivity, plasma shielding effect, and interference from complex matrix effects, which restrict its practical applications [46]. Although LIBS is faster and causes minimal damage to samples, it still has room for improvement compared to traditional detection techniques such as inductively coupled plasma mass spectrometry and atomic absorption spectrometry. In recent years, several studies have emerged aiming to enhance the performance of LIBS, involving both algorithm optimization and modifications to the technique itself. Some researchers have focused on optimizing the algorithm to improve the effectiveness of LIBS detection. For example, Meng et al. [47] incorporated artificial neural networks (ANN) into the detection of copper in soil, which led to improved accuracy and stability of the analytical results. Additionally, modified LIBS techniques that utilize nanoparticles, metal substrates, and plasma confinement have been explored. However, there is currently limited research on the application of these improved LIBS techniques in the detection of fruit and vegetable components. It is expected that with continued optimization, the application of LIBS in the quantitative detection of fruits and vegetables will expand in the future.
Figure 3Laser-induced breakdown spectroscopy system scheme [48].
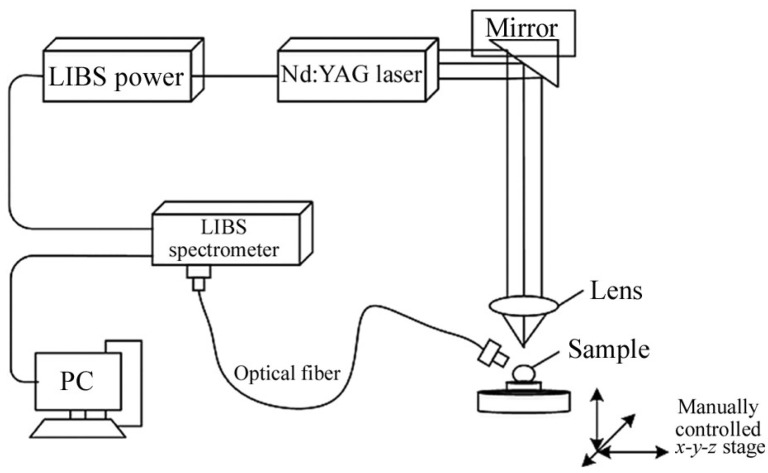



### 3.5. Nuclear Magnetic Resonance (NMR)

NMR is an important technique that utilizes the magnetic properties of atomic nuclei to study the structure, dynamics, and interactions of compounds without damaging the sample. Currently, NMR techniques are categorized into various types depending on the object to be measured, including 1H, 13C, 19F, and 31P. Since hydrogen atoms are widely present in many compounds, 1H spectroscopy is one of the most widely used quantitative NMR methods. The nucleus of an atom with a fixed magnetic moment absorbs electromagnetic waves of a specific frequency in the presence of a magnetic field, jumps from a low energy level to a high energy level, and generates a detection signal. By analyzing the NMR spectra formed by these signals, the structure, dynamics, and interactions of the compounds in the sample can be understood. This technique can be divided into two categories: low-field NMR technology and high-field NMR technology. Low-field NMR detection technology utilizes a magnetic field strength of less than 0.5 T. The instruments used for this method are affordable and widely employed in practical applications due to the lower magnetic field strength. In contrast, high-field NMR detection technology refers to nuclear magnetic resonance conducted with a magnetic field strength exceeding 1.5 T. The higher magnetic field strength in this method results in a superior signal-to-noise ratio, increasing the accuracy and sensitivity of the measurements. Therefore, depending on the desired application and equipment budget, researchers can choose between low-field NMR or high-field NMR for their studies. Both techniques provide insight into and identification of the structure of a sample and play an important role in various scientific fields. NMR technology is widely used in various industries such as agriculture, energy, medical treatment, and food safety due to its simplicity, minimal sample damage, rapid detection speed, and its ability to provide quantitative and qualitative analysis when combined with chemometrics [49,50]. Figure 4 illustrates a simplified schematic of the NMR system. The underlying principle of NMR technology involves the application of a radio frequency (RF) excitation source to administer a specific frequency RF signal to the sample. When the sample is exposed to the RF signal, the hydrogen nuclei within undergo NMR phenomena and absorb energy. Subsequently, the nuclei in the sample relax at a specific frequency and amplitude, releasing energy. By measuring the relaxation time width of each nucleus, it becomes possible to calculate the concentration of each nucleus and the content of other chemical components in the sample [51,52,53].

Despite its advantages, the application of the NMR technique for component quantification is limited due to certain drawbacks, such as low resolution and high detection environmental requirements. These limitations impact the accuracy of detection. In recent years, researchers have been working on addressing these limitations by integrating low-field NMR with near-infrared spectroscopy techniques for joint analysis. For instance, Meng et al. [54] distinguished the origins of various teas by merging NMR and NIR techniques, integrating the spectral data from both methods to create novel NMR-NIR spectra. These combined spectra were then analyzed using chemometric approaches. The findings revealed that the accuracy of results derived from analyzing NMR-NIR data ranged from 86.2% to 95.8%, surpassing the accuracy achieved with NMR alone (68.2% to 78.7%) and NIR alone (80.0% to 89.3%). It is anticipated that further advancements in this area of research will provide a new direction for enhanced detection of components in fruits and vegetables through the integration of NMR with NIR.

### 3.6. Terahertz Spectroscopy (THz)

THz, which operates in the frequency range of 0.1 terahertz to 10 terahertz, mainly detects slow vibrations of molecules and is a technique used for substance detection. It is intermediate between microwave and infrared light on the electromagnetic spectrum, corresponding to wavelengths of 30 µm to 3000 µm. An early terahertz spectrometer was the time-domain spectrometer (THz-TDs). The schematic of the device for THz-TDs is shown in Figure 5. The ultrafast pulse is generated by a femtosecond laser and is divided into a pump beam and a probe beam by a beam splitter. The pumped beam excites the terahertz emitter to produce a terahertz time-domain pulse, which is collimated through a parabolic mirror and focused on the sample. The terahertz pulse carrying the sample information is again collimated and focused onto the terahertz detector. The probe beam is collinear with the terahertz beam and is used to control the gate detector and measure the instantaneous terahertz electric field. The delay system adjusts the time delay between the pump beam and the probe beam and allows iterative sampling of the terahertz timeline. By scanning the time delay, the time-domain waveform of the terahertz pulse is obtained, and the processed terahertz spectral data are visualized in a spectrum diagram. Through meticulous analysis of the terahertz spectrogram and assessment of the absorption peak’s intensity, specific molecular structures present in the sample can be inferred, thereby elucidating the possible components [55,56,57].

In the years leading up to the 1980s, the limited availability of instruments for manipulating and studying terahertz waves led to a heavy reliance on microwave and infrared spectroscopy in electromagnetic wave research. However, with advancements in ultrafast femtosecond lasers and terahertz detectors, the field of terahertz spectroscopy started gaining traction. Researchers began delving into the unexplored realms of terahertz spectroscopy, leading to a gradual increase in the application of this technology [59,60,61]. Terahertz spectroscopy offers numerous advantages [62,63]. THz produces minimal energy release during the detection process, making nondestructive sample detection possible. Compared to other detection techniques, terahertz waves have a high penetration capability and relatively low attenuation after penetration. THz produces unique fingerprint spectra for different samples, which can reveal physical and chemical information inside the sample and be used for qualitative and quantitative analysis of the samples.

Despite its delayed initiation, terahertz spectroscopy is progressively evolving into a promising analytical technique in the agricultural field owing to its outstanding penetration capabilities, rich spectral information, swift detection, and minimal sample damage.

## 4. Application of Rapid and Intelligent Detection Technology

Most advanced detection technologies integrate spectral and frequency data with chemometrics and machine learning to enable intelligent detection of fruit and vegetable components. The detection process is outlined in Figure 6. This comprehensive analysis typically comprises two primary stages: data correction and prediction. The solid-line segment represents the data correction phase, which aims to establish functional relationships. Initially, spectral or frequency data of fruits and vegetables are captured by the device, and their compositional details are determined using conventional analytical methods. Subsequently, chemometrics is applied to remove noise from the spectra or frequency data and correlate it with compositional data to construct an analytical model. The dotted-line segment illustrates the prediction phase. Here, information about the fruit and vegetable under assessment is collected, processed, and fed into the model. Once the model completes its analysis, it facilitates swift and intelligent detection of fruit and vegetable composition [64,65].

### 4.1. Near-Infrared Spectroscopy (NIR)

In recent years, NIR spectroscopy has gained significant attention as a promising approach for various research applications due to its affordability, ease of use, and real-time capabilities. Researchers from both domestic and international backgrounds have successfully utilized this technique, combined with chemometrics, to establish prediction models for the qualitative and quantitative detection of internal constituents in fruits and vegetables. To provide a comprehensive overview of the applications of NIR spectroscopy detection, we have summarized the findings in Table 1.

#### 4.1.1. Health-Promoting Components

(NIR) spectroscopy detection technology is widely used in analyzing the nutrient composition of fruits and vegetables as an indirect detection method. This technology utilizes NIR spectral features and various algorithms to accurately identify the nutrients in agricultural products. As a result, it is an effective tool for assessing the nutritional value of these produce items.

Wang et al. [66] developed an analytical model for the detection of vitamin C content in blueberries using NIR spectroscopy. They compared different preprocessing techniques for spectral data and used competitive adaptive reweighted sampling (CARS) and Random Frog (RF) for variable selection. They found that the multiplicative scatter correction combined with the second derivative (MSC + 2-DER) preprocessing approach was the most suitable for handling the raw spectra of blueberries, and can effectively remove the interfering information in the spectra. Combining this approach with partial least square regression (PLSR), they established a model that provided a theoretical foundation and data support for the intelligent and rapid detection of vitamin C contents in blueberries. In a similar study, Liu et al. [67] developed an analytical model for total flavonoids and anthocyanins content in blueberries by using several methods to preprocess the spectral data, combined with PLSR. Their research served as a basis for monitoring the nutritional quality of fruits and vegetables. Sahamishirazi et al. [68] used NIR for spectral scanning of broccoli, then the scanned broccoli was subjected to chemical indexes, and the processed spectral data, as well as the chemical indexes, were combined to establish a PLSR model to analyze the content of glucosinolates in broccoli. Their model exhibited exceptional performance in determining glucosinolates, enabling the rapid detection of glucosinolates without the need for high-performance liquid chromatography (HPLC) analysis. Pedro et al. [69] developed a model that correlated the chemical constituents of tomatoes, such as lycopene and β-carotene, with the characteristics of tomato near-infrared spectra. They employed preprocessing techniques such as MSC + 2-DER to process the spectra and then the characteristic wavelengths will be extracted from the data to obtain the optimal detection model using PLS-1 modeling. This study further highlights the feasibility of NIR spectroscopy for detecting low-content components in fruits and vegetables and its practical application in the quality sorting of fruits and vegetables by Unilever Brazil.

#### 4.1.2. Harmful Components

The use of calcium carbide has been banned in the artificial ripening of fruits as it contains traces of arsenic. Lakade et al. [70] utilized NIR spectroscopy along with partial least squares (PLS) to detect the presence of As content in mangoes and determine whether calcium carbide was used in the ripening process. This study employed unsupervised techniques, such as principal component analysis (PCA), to differentiate between naturally ripened and artificially ripened mango samples based on spectral data. Subsequently, successive projections algorithm (SPA) was used to extract the characteristic wavelength of the spectrum, and PLS was used to model the As content in mango, so as to detect the use of calcium carbide in the ripening process. This methodology is crucial in preventing the introduction of toxic substances into the market through fruits. In a study by Jamshidi et al. [71], software based on Vis-NIR spectroscopy techniques was developed. The researchers utilized various preprocessing methods to denoise and correct the spectra, incorporating physicochemical data. PLS models and partial least squares-discriminant analysis (PLS-DA) models were established to predict the diazinon content in cucumbers and categorize cucumbers based on diazinon residue. Xue et al. [72] employed Vis-NIR spectroscopy to identify pesticide residue concentrations in navel oranges. They utilized PLS in conjunction with the particle swarm optimization (PSO) algorithm for variable selection and model optimization. Their research confirmed the feasibility of quantitatively detecting dichlorvos in navel oranges using variable selection Vis-NIR spectroscopy and chemometrics.

#### 4.1.3. Components Required for Fruit and Vegetable Growth

Nitrogen (N), phosphorus (P), magnesium (Mg), and other low-content components play a pivotal role in the growth of fruits and vegetables. These components actively contribute to the construction of proteins and chlorophyll formation in the body of fruits and vegetables, thereby promoting their growth and development. Additionally, they modulate the physiological activities of plants, showcasing significant importance in the context of fruits and vegetables.

Shi et al. [73] developed a detection model using NIR spectroscopy to analyze the levels of N and Mg in cucumbers. They optimized the model by determining the number of principal component factors and the K value using a combination of the K-nearest neighbor (KNN) method and genetic algorithm (GA) interactive validation. The NIR spectral data were preprocessed and the total number of subintervals were optimized to obtain the best GA-KNN model. This study demonstrated the feasibility of using NIR spectroscopy for rapid detection of N and Mg in cucumbers. In a separate study, Petisco et al. [74] utilized NIR reflectance spectroscopy to detect the content of N, P, and Ca in 18 woody plants. The spectral data underwent preprocessing using the first derivative (1-DER) and second derivative (2-DER) methods in order to establish correction equations. The models developed through PLSR and multiple linear regression (MLR) were compared, and PLSR was found to exhibit superior accuracy and effectiveness in detecting the levels of Ca in the woody plants.

NIR spectroscopy has proven to be a well-established technique for detecting the composition of fruits and vegetables, as evidenced by numerous studies. However, one challenge in analyzing NIR spectra is the presence of interference from environmental factors and the samples themselves. In order to address this issue, Jiao et al. [75] proposed a systematic evaluation framework for quantifying the impact of preprocessing and identifying an appropriate de-interference approach for multiple datasets. Nevertheless, since the effectiveness of preprocessing depends on the specific data, there is no universal solution available. It is important to note that during spectrum preprocessing, the removal of useful information may lead to an increase in detection errors. To overcome this, Abrahamsson et al. [76] introduced a method based on time-resolved spectroscopy and diffusion theory to correct spectral data, which improved the predictive power of the method by 50% compared to a model solely based on traditional NIR data. Moreover, in addition to addressing the linearity aspect of spectra, some researchers are exploring ways to enhance NIR spectra from a nonlinear perspective. Lv [77] and others proposed a denoising stacked autoencoder model that effectively extracts low-dimensional feature information from NIR spectra by evaluating robust samples. These advancements in NIR spectroscopy show great potential for enhancing its application in component detection.

### 4.2. Hyperspectral Imaging (HSI)

HSI technology is a cutting-edge form of intelligent detection that combines traditional imaging technology with spectroscopic technology. This integration allows for the precise, efficient, and nondestructive collection of physiological data pertaining to fruits and vegetables, covering aspects such as growth patterns, pathological characteristics, and quality attributes.

In recent years, there has been a shift in the focus of HSI technology research from qualitative to quantitative analysis of fruits and vegetables. This shift is evident from the growing body of literature exploring the application of HSI technology to accurately measure the composition of these agricultural products (Table 2) [78]. This emerging research trend represents a significant advancement in the field of HSI technology, as it provides a more precise and objective approach to analyzing the composition of fruits and vegetables.

#### 4.2.1. Health-Promoting Components

Assessment of the nutritional value of fruits and vegetables relies heavily on their low-content composition. In a study conducted by Malegori et al. [79], ten acerolas were examined at different ripening stages (green, yellow, and red) based on color. Hyperspectral imaging (NIR-HSI) and Fourier transform near-infrared spectroscopy (FT-NIR) were used to obtain information about the samples. Preprocessing techniques such as standard normal variate (SNV) and Savitzky–Golay (SG) were applied to ensure consistency in the spectral data. The classical least squares (CLS) model was utilized to identify the presence of vitamin C by comparing the spectra of the samples with the reference spectra (samples containing different concentrations of vitamin C) and to generate the corresponding distribution maps, which showed that the vitamin C content in the different fruit regions in the needles gradually decreased during the maturation process. Similarly, Guo et al. [80] developed a model for detecting vitamin C content in potatoes using Fisher’s discriminant analysis (FDA) and data fusion. Various preprocessing methods were compared, and multiple scattering correction (MSC) was found to be the optimal approach for potato spectral data. Competitive adaptive reweighted sampling (CARS) was used for variable selection and reducing redundant information in the spectra. The processed data were combined with FDA to establish a backpropagation neural network (BPNN) model to predict the range of vitamin C. Fusion of FDA with the spectral data at the extracted characteristic wavelengths to form a new variable can reduce the effect of correlation between variables on the model, which was used in the BPNN model to improve the prediction accuracy of the content of vitamin C in potatoes. In another study, Chen et al. [81] utilized near-infrared hyperspectral imaging and the RBF-PLS model to rapidly detect pomelo quality. Partial least squares (PLS) is a multivariate linear regression analysis method, which is not as effective as nonlinear regression in detecting complex objects when facing them. The authors improved PLS by invoking the nonlinear kernel function Gaussian radial basis function (RBF) as an algorithmic embedding for PLS and optimized the modeling process parameters. The compositional data in pomelo were input into the RBF-PLS model for model training, the model was detected with unknown samples, and the prediction results were also quite impressive. The experimental results show that the proposed RBF-PLS model combined with near-infrared hyperspectral imaging technology is feasible to quantitatively detect the target content of pomelo fruits. They provide valuable guidance for future developments in this field.

#### 4.2.2. Chlorophyll

Chlorophyll is an essential component in the process of photosynthesis, and its concentration in plants typically ranges from 0.5 to 3 mg/g. Plants that have a deficiency in chlorophyll find it challenging to efficiently utilize sunlight for nutrient replenishment, resulting in slow growth and reduced yields [87].

Kjær et al. [82] used HSI techniques to predict the concentration of chlorophyll in potatoes. They conducted an experiment with four different potato varieties subjected to various treatments and light conditions to induce changes in the relative content of chlorophyll. The same model was then utilized to detect chlorophyll content in the potatoes. The results demonstrated consistent responses of the different potato varieties to the treatments and light conditions, with R^2^ values ranging from 0.93 to 0.97. This method shows promise for predicting chlorophyll content in potatoes. Chlorophyll content in plants is closely related to nitrogen content, a key component of chlorophyll. Shi et al. [83] collected spectral data of multiple leaves from nitrogen-deficient and normal plants, respectively, using hyperspectral imaging techniques. They then determined the accurate chlorophyll content data in the leaves using high-performance liquid chromatography (HPLC), solved the problem of scattering interference in the estimation of chlorophyll concentration in cucumber by using standard normal variables (SNV), and established a calibration model to correlate the spectra with the chlorophyll concentration determined by HPLC. By combining the model with near-infrared hyperspectral image data to calculate the chlorophyll distribution map on cucumber leaves, the obvious difference between nitrogen-deficient leaves and control leaves could be directly observed. This approach helps overcome the problem of uneven nutrient distribution in fruits and vegetables, thereby enhancing overall yield. In another study, Sun et al.’s [84] investigation determined that chlorophyll content decreases significantly when peach is infected by pathogens. Thus, chlorophyll content was detected by using HSI and honey peach rot was recognized based on the chlorophyll level. In this study, three optimal wavelengths (617 nm, 675 nm, and 818 nm) were selected by successive projections algorithm (SPA) to build quantitative partial least squares (PLS) and SPA-PLS models for chlorophyll content for determining chlorophyll content. For classification, although it is possible to classify diseased peaches based on complete spectral data, the dataset is too large for practical application, so the team obtained three band ratios (B617/818, B675/818, and B675/617) from the spectral data to synthesize images for distinguishing early-stage diseased peaches from healthy peaches, and demonstrated an accuracy of 98.75%. This study represents a pioneering use of HSI combined with chlorophyll content for disease detection in fruits and vegetables. The results highlight the potential of HSI to accurately assess chlorophyll content and classify decayed peaches based on chlorophyll levels, providing a novel method for quality detection in fruits and vegetables.

#### 4.2.3. Heavy Metal

With the worsening of environmental pollution, the presence of heavy metal elements in fruits and vegetables is on the rise, as they absorb these elements from the air, water, and soil during their growth process. This poses a health risk for residents who consume fruits and vegetables that contain excessive levels of heavy metals. For instance, heavy metal poisoning can occur if an individual consumes over 100 mg of copper in a single instance.

Souza et al. [85] conducted a study on predicting cadmium concentrations in kale and basil using HSI and machine learning. The researchers used traditional methods to determine the cadmium concentrations at the time of collection and then utilized the collected data for machine learning purposes. They also collected and processed Vis/NIR images. To select the most relevant variables, the reflectance spectra underwent variable selection through RF before being applied to three machine learning models. The results showed that the artificial neural network (ANN) was effective in detecting cadmium content in leaves. The ANN successfully classified the samples using a threshold of 0.2 mg/kg to differentiate between samples with cadmium concentration exceeding the limit and those below it. In a similar study, Zhou et al. [86] used fluorescence hyperspectral imaging to detect cadmium content in lettuce leaves. They applied five different spectral preprocessing algorithms (Savitzky–Golay, SG; multiple scattering correction, MSC; standard normal variate, SNV; first derivative, 1-DER; second derivative, 2-DER) to process the spectra. After determining the optimal preprocessing method, the researchers performed dimensionality reduction using various variable selection methods (successive projections algorithm, SPA; competitive adaptive reweighted sampling, CARS; iteratively retaining informative variables, IRIV; variable iterative space shrinkage approach, VISSA; wavelet transform combined with stepwise regression, WT-SR). The processed data were then fed into a multiple linear regression (MLR) model to establish a prediction model for cadmium content in lettuce leaves. The MLR model using the 1-Der + WT-ST algorithm was found to be the most effective in predicting cadmium content and visualizing the distribution of heavy metals in lettuce leaves.

### 4.3. Raman Spectroscopy

Compared to conventional detection techniques, Raman spectroscopy offers several advantages in terms of sensitivity, resistance to water and background interference, simplicity in sample preprocessing, and rapid nondestructive detection. These unique features have led to the widespread application of Raman spectroscopy in the field of food safety. Table 3 presents the extensive research conducted by various countries and research teams on the use of Raman spectroscopy for detecting trace substances in fruits and vegetables.

#### 4.3.1. Mycotoxin

During the growth and storage processes of fruits and vegetables, environmental factors such as temperature and humidity make them vulnerable to fungal growth. Common fungal species that affect these crops include Aspergillus, Fusarium, and molds. Mycotoxins are metabolites produced by fungi, and human ingestion of mycotoxins may lead to acute food poisoning, liver and immune system damage, and carcinogenic risk. One notorious example is aflatoxin, which, when consumed excessively, can lead to various health issues including liver damage, immune system suppression, and potential carcinogenic effects. Therefore, food safety standards have strict limits on aflatoxin levels. In Chinese food safety standards, the maximum allowable level of aflatoxin in daily food for residents is specified as 0.5–20 μg/kg [94,95].

Gabbitas et al. [88] developed a rapid method using label-free enhanced Raman spectroscopy (SERS) with colloidal gold nanoparticles for simultaneous detection of aflatoxin B1 (AFB1), zearalenone (ZEN), and ochratoxin A (OTA) in maize. Reserve solutions of AFB1, ZEN, and OTA were mixed with SERS nanosubstrates and subsequently placed on corn for data acquisition. After applying a second-order guide to the SERS spectrum to eliminate baseline bias and isolate overlapping peaks, each mycotoxin had its unique Raman fingerprint, which could be clearly distinguished by principal component analysis. The study also evaluated the quantitative power of the method to determine whether concentrations of multiple mycotoxins in maize could be predicted. Different concentrations of mycotoxins in maize would cause the difference in peak intensity of the corresponding SERS spectrum, so it could be used for quantitative analysis. The linear relationship between predicted concentrations and actual concentrations was evaluated based on SERS spectra of known concentrations combined with partial least squares regression models with concentrations up to 1.5 ppm (4.8 μM) for AFB1, ZEN, and OTA, with correlation coefficients of 0.74, 0.89, and 0.72, respectively. Guo et al. [89] established a high-throughput label-free detection model and synthesized AuNRs substrate with a good enhancement effect. The coffee ring structure with a regular shape and good enrichment effect was optimized by comparing different drip volumes and drying temperatures, and the sensitivity of detection was improved. After pretreatment and feature extraction of the SERS spectrum, Patulin (PAT) and alternariol (AOH) in apples were quantitatively analyzed by the stoichiometric method. Partial least squares with a genetic algorithm (GA-PLS) model showed the best performance for AOH, Rp = 0.9759, RMSEP = 0.336, and the joint interval partial least squares (SiPLS) model showed the best performance for PAT, Rp = 0.9759, and RMSEP = 0.378. The detection limit of PAT and AOH was reduced to 1 μg L-1. The rapid detection of PAT and AOH toxins in apples was realized, and the marking-free detection could improve the detection efficiency and reduce the cost. Lee et al. [90] conducted a study combining Raman spectroscopy with chemometrics to quantitatively and categorically analyze aflatoxin contamination in maize. Linear discriminant analysis (LDA) was used to classify corn samples with and without aflatoxin, achieving correct classification rates ranging from 94 to 100%. Among the models developed for predicting aflatoxin concentrations, the partial least squares regression (PLSR) model exhibited the best detection accuracy, performing comparably to the high-performance liquid chromatography (HPLC) reference value.

#### 4.3.2. Pesticide Residue

Huang et al. [91] demonstrated the use of surface-enhanced Raman scattering (SERS) spectroscopy with colloidal gold nanoparticles for the rapid detection of phosalone residues in pakchoi. The entire detection process, including sample extraction and spectral acquisition, was completed in approximately 15 min. Preprocessing of the raw spectra using three methods (multiple scattering correction, MSC, standard normal variate SNV, and normalization) was followed by the development of a partial least squares (PLS) model. The combination of MSC and PLS showed the best detection performance. To validate the accuracy of the model, five paraquat samples with unknown phosalone concentrations were analyzed. The predicted values from the model were in agreement with the measured values obtained using gas chromatography–mass spectrometry (GC-MS), with no significant difference observed. Thus, this method proves effective for the rapid detection of phosalone pesticide residues in pakchoi. Xie et al. [92] introduced a novel SERS method for the swift quantitative and qualitative detection of methamidophos in vegetables. The study compared different experimental conditions related to solvents and pH values. An orthogonal experimental design was used to determine methamidophos contents in six vegetable samples. The standard curves showed excellent linearity within the range of 0.1~100 μg/mL. The relative standard deviations (RSD) were within the range of 1.2–2.5%, and the detection limit was 0.01 μg/mL. These results highlight the capability of surface-enhanced Raman spectroscopy as a robust tool for detecting methamidophos in fruits and vegetables. The method has potential applications in detecting other food contaminants to ensure food safety. Yaseen et al. [93] developed a silver-coated gold nanoparticle-based SERS method for the rapid detection of multiple organophosphorus chemical pesticides in peach fruits. The study compared the Raman spectra of silver-coated gold nanoparticles with those of single silver nanoparticles and single gold nanoparticles. The silver-coated gold nanoparticles proved to be more suitable for detecting organophosphorus compounds in various samples. The method showed high recoveries ranging from 93.36% to 123.6% and could be applied for the analysis of trace contaminants like triazophos and methyl-parathion in different food matrices.

### 4.4. Laser-Induced Breakdown Spectroscopy (LIBS)

LIBS is a real-time, non-invasive method for analyzing multi-element samples without the need for sample pretreatment. By adjusting the irradiation conditions such as laser wavelength, pulse duration, pulse energy, and focusing geometry, LIBS can be applied to analyze various substances. The continuous advancement of LIBS technology has enabled more accurate and convenient analysis of low-content components in fruits and vegetables [96]. Table 4 provides examples of the applications of LIBS.

#### 4.4.1. Health-Promoting Components

Rai et al. [97] conducted a study on the application of LIBS technology in the detection of trace elements such as Na, K, Mg, and Ca in momordica charantia. They did this by giving different doses of bitter melon frozen fruit powder to animals in animal experiments and observing its effects on the animals’ blood sugar levels. By monitoring and comparing the blood glucose levels of different experimental groups, the author can explore the relationship between the trace elements in bitter melon and the changes in blood glucose levels. In addition to LIBS, the researchers also utilized atomic absorption spectroscopy (AAS) to analyze the low-content components in Momordica charantia. The results showed a consistent measurement of low-content components between LIBS and AAS, demonstrating the capability of LIBS in detecting low-content components. In another study, Singh et al. [98] quantitatively analyzed Na, K, Mg, Ca, and other trace elements in melon seeds using LIBS combined with the calibration curve method. The results obtained from LIBS were found to be in agreement with the findings from AAS. This study not only confirmed that melon seeds are rich in essential nutrients for the human body but also demonstrated the feasibility of employing LIBS technology for detecting low-content components in fruits and vegetables.

#### 4.4.2. Pesticide Residue

Wu et al. [99] applied LIBS and HSI techniques for the rapid detection of thiophanate-methyl residue in mulberry fruits. They utilized the competitive adaptive weighted sampling (CARS) algorithm to simplify spectral data after preprocessing. This approach was combined with principal component analysis (PCA) and partial least squares regression (PLSR) modeling to both qualitatively and quantitatively analyze thiophanate-methyl residue in a variety of mulberry fruit samples. In a similar vein, Zhao et al. [100] enhanced the detection of various fruits and vegetables using metal nanoparticles to amplify the signal of LIBS. By analyzing characteristic peaks in LIBS spectra, they were able to determine the content of P, S, and Cl, leading to the quantification of pesticide residues in these products. The use of metal nanoparticle-enhanced LIBS significantly lowered the detection limits of pesticide residues compared to standard LIBS, indicating its potential for precise quantification in fruits and vegetables. Martino et al. [101] investigated the application of the LIBS technique for the rapid measurement of pesticide residues in chard leaves. They employed spectral data preprocessing to identify the contents of P, S, C, and Cl based on LIBS spectra. Classification through PCA, followed by validation using linear discriminant analysis (LDA), yielded a detection error rate of less than 9.5% in their analysis. Overall, these studies highlight the effectiveness of LIBS and the potential to enhance its capabilities through techniques such as nanoparticle amplification and data preprocessing. The combination of LIBS with various analytical approaches shows promise for the accurate and rapid detection of pesticide residues in fruits and vegetables.

#### 4.4.3. Heavy Metal

Detecting heavy metals in fruits and vegetables is crucial for ensuring food safety. Researchers have explored the use of LIBS for this purpose. Shen et al. [102] employed LIBS to analyze the spectral information of lettuce and detect the presence of Cd. They used various preprocessing techniques in combination with genetic algorithms (GA) to select 22 variables. Based on these variables, they established a PLS model for accurate Cd content detection. In addition to LIBS, K-nearest neighbor (KNN) and random forest (RF) algorithms were utilized to assess the level of Cd contamination. The study demonstrated that LIBS is a rapid and effective method for evaluating heavy metal contamination. Building upon LIBS, Yang et al. [103] proposed a novel analytical framework for detecting Cu and Cd contents in mulberry leaves. Their framework incorporated Self-organizing Mapping (SOM), as well as variable selection methods such as the successive projections algorithm (SPA) and Uninformative Variable Elimination (UVE). By combining these techniques with a PLS model, they achieved high prediction accuracies for copper and chromium content in mulberry leaves. The best model obtained Relative Prediction Deviation (RPD) values of 10.0494 and 8.3874 for copper and chromium content, respectively. The Root Mean Square Error of Prediction (RMSEP) for copper and chromium content reached 110.4550 and 41.4561, respectively. This methodology not only reduces data complexity but also improves model accuracy, making it highly relevant for detecting heavy metals in fruits and vegetables.

### 4.5. Nuclear Magnetic Resonance (NMR)

NMR is a powerful tool for detecting and identifying different substances based on their signal characteristics. It has been widely utilized in various applications, such as moisture and fat detection, identifying food adulteration, and assessing meat quality. For example, Xu et al. [104] used NMR and magnetic resonance imaging (MRI) to investigate moisture distribution in broccoli tissues. Hatzakis et al. [105] employed NMR to quantify free glycerol in virgin olive oils from different Greek regions. Siciliano et al. [106] utilized high-resolution H^−1^ NMR spectroscopy to determine fatty acid chain distribution in maturing pork products. However, when it comes to the detection of low-content components in fruits and vegetables, there is limited research in this area, resulting in fewer applications of NMR techniques. Therefore, further studies are needed to explore the potential of NMR in this domain.

Capitani et al. [107] conducted a study to evaluate the internal condition of Hayward kiwifruits using a portable unilateral NMR instrument. They employed high-field NMR spectroscopy to track the changes in amino acids, sugars, organic acids, and other metabolites within the kiwifruits from June to December. This allowed them to gain insights into the variations in nutrient content during different growth stages and determine the optimal time for harvest. In a similar vein, Clausen et al. [108] utilized a metabolomics approach based on 1H NMR to analyze the differences in sugars and β-carotene content among various carrot genotypes. They employed the NMR technique in conjunction with principal component analysis (PCA) to identify compositional variations between the genotypes. This methodology not only enables the investigation of the authenticity of plant foods but also facilitates the assessment of the impact of post-harvest handling and storage on the quality of plant foods. Ultimately, this research contributes to improving the overall quality and diversity of plant-based products available in the market.

NMR, with its ability to provide detailed information about molecular structure and composition, facilitates rapid, intelligent, and nondestructive detection of fruits and vegetables, holding significant potential for future development.

### 4.6. Terahertz Spectroscopy (THz)

THz has gained significant attention in various fields such as biomedicine, wireless communications, food safety, and agriculture due to its low energy, exceptional penetration, and nondestructive detection capabilities [109]. Notably, advancements in THz have been made in biomedicine and wireless communication. In biomedicine, it is beneficial for determining and localizing tumors, while in wireless communication, researchers have explored its application in 6G systems. Although the use of THz in food safety and agriculture is still in its early stages due to high research costs, it shows potential for detecting food adulteration and additives and classifying genetically modified plants [110,111,112,113]. In food safety, THz is effective for identifying food adulteration and additives, while in agriculture, it aids in detecting crop moisture and sugar content [114,115]. However, the exploration of low-content components in fruits and vegetables remains an ongoing area of study.

One of the broader applications of THz technology is its potential to detect low-content components, including pesticide residues. Baek et al. [116] used terahertz time-domain spectroscopy (THz-TDs) to quickly detect the pesticide methomyl in food. However, they faced challenges due to the technology’s low sensitivity and the limited quantity of pesticide residues in food, which made detecting them difficult. To address these issues, several researchers proposed targeted studies. Qi et al. [117] and Qin et al. [118] enhanced the terahertz spectroscopy system structure by incorporating metamaterial sensors based on single and double U-shaped resonators, as well as metal ohmic ring arrays. These modifications aimed to improve the sensitivity of the terahertz detection technique and enable the realistic detection of pesticide residues. Although these sensitivity enhancements were achieved, the traditional THz-TDs method analyzed absorption peaks in the spectra to determine pesticide residue concentrations. This approach could overlook other essential information about the pesticides, resulting in detected concentrations lower than the actual values. To overcome this limitation, Chen et al. [119] utilized the asymmetric least squares (ASLS) method to calibrate the terahertz spectra, improving the signal-to-noise ratio. They combined this method with chemometric techniques such as partial least squares (PLS), support vector regression (SVR), interval partial least squares (iPLS), and backward interval partial least squares (biPLS) to successfully detect imidacloprid in rice samples. In another study, Ma et al. [120] employed spectral preprocessing and genetic algorithms (GA) to enhance the accuracy of a backpropagation neural network (BPNN) model. This approach, coupled with THz technology, facilitated high-precision quantitative analysis of low-concentration ternary blends of pesticides in wheat flour. These findings demonstrate the feasibility of combining terahertz spectroscopy and chemometrics for the detection of pesticide residues, thereby expanding the application of THz technology in detecting low-content components.

## 5. Conclusions

This paper provides an overview of the intelligent and rapid detection technologies that have gained widespread adoption in recent years. It aims to outline the advantages and disadvantages of these technologies, as summarized in Table 5. These advanced techniques effectively address the limitations of traditional detection methods, such as prolonged detection times, complex preprocessing procedures, and the potential for environmental pollution. By integrating chemometrics, sensors, and other cutting-edge technologies, these methods enable intelligent and swift detection of low-content components in fruits and vegetables. They not only improve the efficiency of detecting internal nutrients and potentially harmful substances but also play a crucial role in monitoring the changing condition of fruits and vegetables throughout their growth, transportation, and storage. The application of these technologies has the potential to significantly reduce waste, enhance the quality of fruits and vegetables, and provide consumers with reliable assurance of food quality and safety when purchasing these products.

Among the myriad intelligent detection technologies, NIR detection technology, a relatively mature method for detecting low-content components, has been widely employed in conjunction with chemometric methods for intelligent and rapid detection of fruit and vegetable composition. Portable NIR detection equipment is gaining attention, paving the way for its practical quantitative application. In contrast, HSI, an emerging technology that integrates image and machine learning, has garnered significant interest in agriculture. While capable of obtaining richer spectral information on fruits and vegetables, it is suitable for large-scale field detection, particularly in composition detection during the growth stage. However, its information redundancy and interference characteristics impact detection speed and accuracy. Raman spectroscopy, with the ability to recognize molecular structures and functional groups, holds promise for application in fruit and vegetable composition detection. Challenges include low sensitivity, susceptibility to interference from other components, and high costs. Ongoing research aims to enhance signal intensity and reduce costs to improve the technique’s detection capability. LIBS, despite limitations in sensitivity and measurement accuracy, offers the unique advantage of simultaneously analyzing multiple elements and effectively analyzing samples of different material forms. Current research focuses on optimizing algorithms and detection instrument structures to further enhance the technique.

The NMR and THz techniques have the potential to greatly enhance the efficiency of data processing and analysis in assessing the quality and safety of fruits and vegetables. These techniques are known for their rapid detection speeds and nondestructive properties. However, their broader application in detecting low-content components in fruits and vegetables has been limited due to the high instrument costs and sensor sensitivity constraints. As a result, there is a scarcity of reported studies in this area.

### 5.1. Existing Problems

The seven intelligent detection techniques mentioned above provide a solid theoretical basis for the intelligent rapid detection of low-content components in fruits and vegetables. However, the following problems still need to be solved in order to successfully apply these intelligent rapid detection techniques to field analysis:Sensor stability problems are hindering the accurate detection of low-content components in fruits and vegetables. Despite being crucial components of detection devices, sensors face challenges relating to their stability. Firstly, environmental factors significantly affect sensor performance. For instance, temperature fluctuations lead to temperature drift errors, resulting in inconsistent detection results at different temperatures. Secondly, traditional sensor materials have limited response capabilities, thus impacting detection sensitivity. While using new materials can mitigate this issue, the complex preparation process and high cost associated with these materials hamper their commercial implementation. Moreover, large-scale production and utilization of these new materials pose additional obstacles. As a result, addressing these challenges requires innovative approaches to improve sensor stability and overcome the limitations of traditional materials.Sample storage and handling methods in the field of rapid detection technology and chemometrics lack standardization, resulting in potential discrepancies in the data obtained from the same sample in different studies. While intelligent rapid detection technology combined with chemometrics has improved detection efficiency and minimized environmental and sample interference, the results are still influenced by sample characteristics such as skin thickness, surface irregularities causing scattering, and internal moisture of fruits and vegetables. Consequently, a significant amount of extraneous information is present in the results, affecting the accuracy of the test results for fruits and vegetables that differ in origin, variety, and transportation conditions. It is crucial to explore methods to mitigate the impact of these characteristics and external conditions. Additionally, when detecting components present in low concentrations, the instrument’s sensitivity and service life can contribute to errors. Addressing this technical challenge necessitates effective improvement in the instrument’s structure to accommodate the detection of low-content components.Algorithmic problems arise in the field of intelligent rapid detection technology combined with chemometrics, particularly when establishing qualitative and quantitative analysis models for the rapid and intelligent detection of low-content components in fruits and vegetables. One major challenge is the reliance on a large amount of sample data to improve and optimize the model, which can lead to information overload and negatively impact detection efficiency. Thus, it is crucial to effectively mine the data in the detection results and establish optimal detection models. Moreover, the current models lack versatility for databases of different samples, necessitating the establishment of separate databases for each type of sample. This can increase the workload of the model and potentially affect its stability and repeatability. Therefore, it is necessary to address the issue of developing a versatile and stable model for fruit and vegetable composition detection, which remains a prominent concern in this field.

### 5.2. Prospects

In the context of the era of intelligence and diversification, as well as the development of intelligent and rapid detection technology, become a trend with great market value and social value.

To enhance the stability of the sensor, various approaches can be considered. Firstly, it is crucial to conduct research on new sensor materials that have minimal environmental impact but can still improve sensitivity. This can significantly contribute to the overall stability of the sensor. Additionally, the implementation of multi-sensor fusion technology can further enhance system stability and improve detection accuracy. Through the collaborative effect of different sensors, the overall performance of the system can be greatly improved. To address the influence of environmental factors on the sensor, it is advisable to integrate advanced environmental compensation technology. Temperature compensation algorithms can be employed to reduce the impact of temperature variations on the sensor’s performance. This would extend its applicability across diverse environmental conditions. To ensure reliable performance, regular testing and maintenance of the sensor are necessary. This practice helps to identify and rectify any issues promptly, ensuring continued stable operation.To optimize the structure of testing instruments, it is crucial to focus on the core components that directly influence testing accuracy. By enhancing the sensitivity of sensors, stabilizing the signal processing system, and increasing the brightness of the light source, the detection accuracy can be significantly improved. These optimization strategies aim to minimize interference factors generated by the instrument, enhance electrical signals captured during the detection process, and ultimately achieve precise detection of low-content components in fruits and vegetables.Optimization of artificial intelligence algorithms: In the process of detecting fruit and vegetable ingredients, artificial intelligence algorithms play a vital role through data processing, feature variable selection, and modeling analysis. The optimization of these artificial intelligence algorithms can better reduce data errors and interference, and improve the accuracy of fruit and vegetable composition detection. The optimization of AI algorithms can start by reducing the redundant information in the data, reducing the data dimension; combining the cross-validation method to verify the model, selecting the best combination of parameters to improve the accuracy of the model; and adopting more complex neural network structure and training strategy to improve the model’s explanatory and accuracy. These optimization directions will make the algorithm process the data more efficiently, and thus improve the reliability and accuracy of fruit and vegetable composition detection.Utilize intelligent rapid detection technology and integrate it into various applications, such as a mobile app, to enable real-time transmission and sharing of data. This would allow users to conveniently view and share test results on their cell phones, providing a timely solution for the detection of fruits and vegetables with low ingredient content. Combining the advantages of intelligent detection technology and mobile app functionality, it offers a more convenient, intelligent, and comprehensive approach to testing the quality of produce.

## Figures and Tables

**Figure 1 foods-13-01116-f001:**
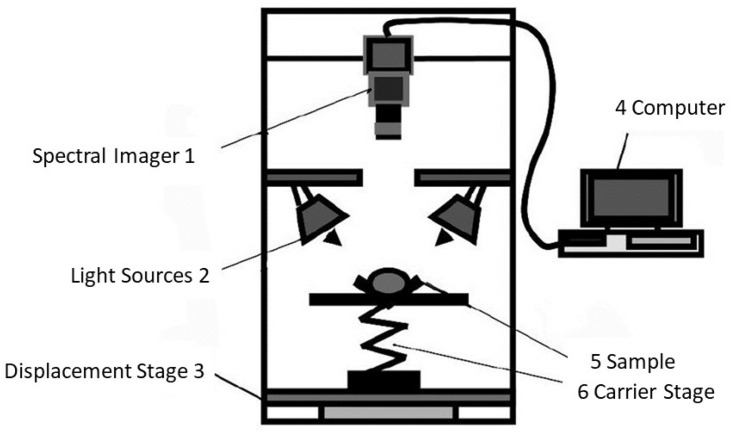
Schematic structure of hyperspectral imaging device [29].

**Figure 4 foods-13-01116-f004:**
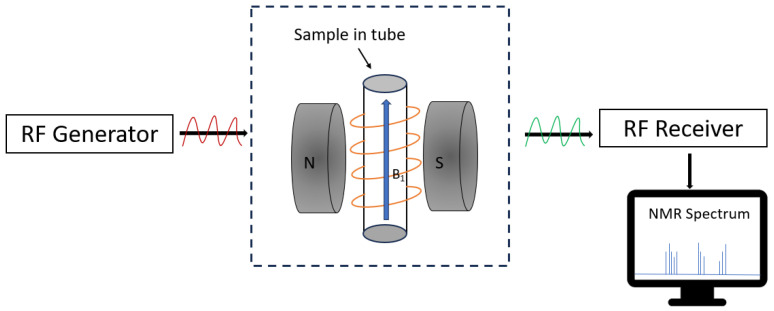
Simplified schematic of an NMR system.

**Figure 5 foods-13-01116-f005:**
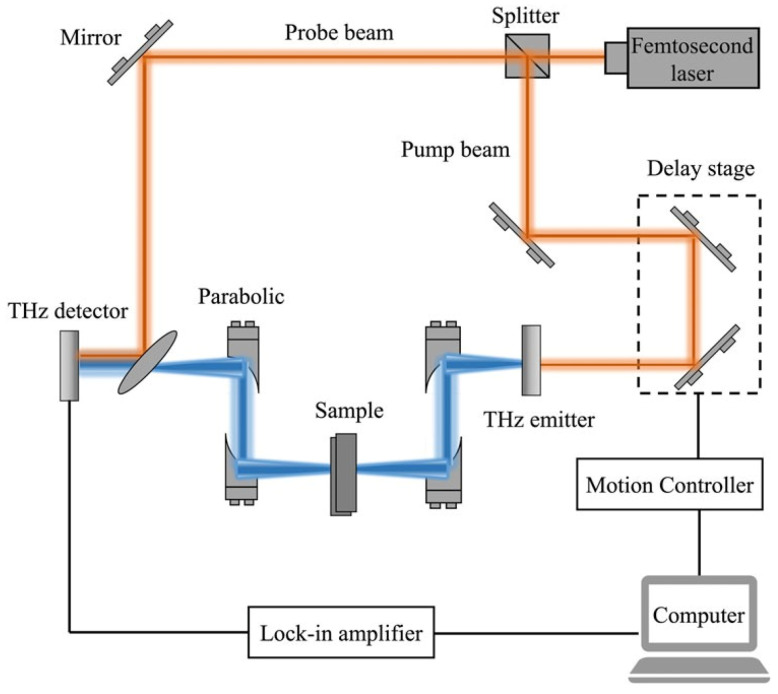
Schematic diagram of the terahertz spectral system [58]. Reprinted with permission from Ref. [58]. 2022, Frontiers.

**Figure 6 foods-13-01116-f006:**
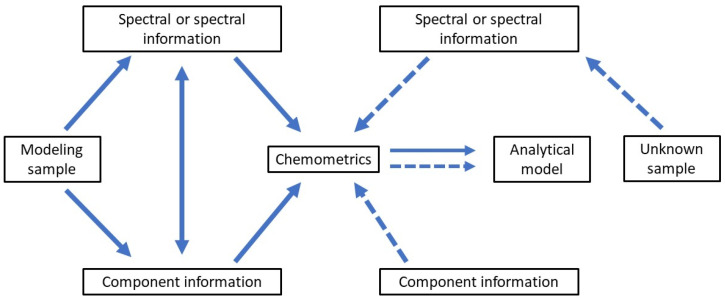
The realization process of detection technology.

**Table 1 foods-13-01116-t001:** Application of NIR spectroscopy to the low-content components of fruits and vegetables.

Target of Detection	Component	Preprocessing	Variable Selecting	Optimal Model	Effect	Reference
Blueberry	SSC, Vitamin C	MSC + 2-DER	CARS + RF	PLSR	SSC: RMSEP = 0.9673 R_P_ = 0.7376 Vitamin C: RMSEP = 3.6885 R_P_ = 0.7021	[66]
Blueberry	Total Flavonoids, Anthocyanins	1-DER	/	PLSR	Total Flavonoids: R_P_^2^ = 0.7968 Anthocyanins: R_P_^2^ = 0.7902	[67]
Broccoli	Glucosinolates	/	/	PLSR	R^2^ = 0.5–0.78 RPD = 1.35–2.19	[68]
Tomato	Lycopene, β-carotene	MSC + 2-DER	/	PLS-1	Lycopene: R_P_^2^ = 0.9996 β-carotene: R_P_^2^ = 0.9981	[69]
Mango	As	/	PCA + SPA	PLS	R^2^ ranged from 0.9 to 0.96 for different arsenic concentrations	[70]
Cucumber	Diazinon Residue	MSC + 1-DER	PSO	PLS-DA	R_CV_ = 0.91 SECV = 3.22	[71]
Navel Oranges	Dichlorvos	/	PSO	PLS	R^2^ = 0.8732	[72]
Cucumber	N, Mg	SNV	GA	KNN	Training Set Recognition Rate: 98% Prediction Set Recognition Rate: 96%	[73]
Woody Plant	N, P, Ca	data	/	PLSR	Ca: R^2^ = 0.91 P: R^2^ = 0.74 N: R^2^ = 0.95	[74]

**Table 2 foods-13-01116-t002:** Application of HSI to the low-content components of fruits and vegetables.

Target of Detection	Component	Preprocessing	Variable Selecting	Optimal Model	Effect	Reference
Acerolas	Vitamin C	SNV+SG	PCA	CLS	The amount of vitamin C gradually decreases during growth.	[79]
Potato	Vitamin C	MSC	CARS	FDA + BPNN	R^2^ = 0.999 REMSP = 0.1727	[80]
Pomelo	Vitamin C	/	/	RBF + PLS	RMSEV = 41.381 mg/kg	[81]
Potato	Chlorophyll	/	/	PLSR	R^2^ = 0.93~0.97	[82]
Cucumber	Chlorophyll	SNV	PCA	MLR	R^2^ = 0.8712	[83]
Honey Peach	Chlorophyll	/	SPA	PLS, SPA-PLS	PLS: R_P_^2^ = 0.904 RMSEP = 0.633% SPA-PLS: R_P_^2^ = 0.858 RMSEP = 0.751%	[84]
Kale and Basil	Cadmium	/	RF+PCA	ANN	Ability to classify plants according to cadmium concentration above or below 0.2 mg/kg.	[85]
Lettuce Leaves	Cadmium	1-DER	WT-ST	MLR	R_P_^2^ = 0.7905 RMSEP = 0.0096	[86]

**Table 3 foods-13-01116-t003:** Application of Raman spectroscopy to low-content components of fruits and vegetables.

Target of Detection	Component	Preprocessing	Variable Selecting	Optimal Model	Effect	Reference
Maize	AFB1, ZEN, and OTA	/	/	PLS	The correlation coefficients for AFB1, ZEN, and OTA were 0.74, 0.89, and 0.72, respectively, by PLS model.	[88]
Apple	PAT, AOH	2-DER	Si, GA	Si-PLS, GA-PLS	PAT: SI-PLS: R_C_ = 0.9905 R_P_ = 0.9759 AOH: GA-PLS: R_C_ = 0.9829 R_P_ = 0.9808	[89]
Maize	Aflatoxin	Deconvolution	/	PLSR	R_V_^2^ = 0.990	[90]
Pakchoi	PHO	MSC	/	PLS	R_P_^2^ = 0.9807 RMSECV = 0.886 mg/L	[91]
Vegetables	MAP	/	/	SPSS	R^2^ = 0.9852	[92]
Peach	Organophosphorus Chemical Pesticides	/	/	/	The detection limit was 0.001 mg/kg.	[93]

**Table 4 foods-13-01116-t004:** Application of LIBS to low-content components of fruits and vegetables.

Target of Detection	Component	Preprocessing	Variable Selecting	Optimal Model	Effect	Reference
Momordica Charantia	Na, K, Mg, Ca	/	/	SPSS	The LIBS technology test matches the AAS test results.	[97]
Melon Seed	Na, K, Mg, Ca	/	/	SPSS	*p* < 0.05	[98]
Mulberry Fruit	Thiophanate-Methyl Residue	SNV	CARS	PLSR	RPD = 2.585 RMSEP = 7.09 × 10^−4^ R_P_^2^ = 0.921	[99]
Fruits and Vegetables	P, S, CI	/	/	/	Detection limits are two orders of magnitude lower than typical detection limits.	[100]
Chard Leaves	P, S, C, Cl	/	PCA	LDA	Detection error rate of less than 9.5%.	[101]
Lettuce	Cd	Normalization	GA	PLS	R_p_^2^ more than 0.94, LODs less than 5.5 mg/kg	[102]
Mulberry Leaves	Cu, Cr	/	SOM, SPA, UVE	PLS	Cu: RPD = 10.0494 RMSEP = 110.4550 Cr: RPD = 8.3874 RMSEP = 41.4561	[103]

**Table 5 foods-13-01116-t005:** Advantages and disadvantages of different intelligent rapid detection technologies.

Detection Technology	Advantage	Disadvantage
Near-infrared Spectroscopy	Fast and inexpensive detection	Vulnerable to environmental disturbances, low stability
Hyperspectral Imaging	More comprehensive spectral data	Excessive volume and complexity of information, requiring time-consuming data analysis
Raman Spectroscopy	Rapid, accurate, nondestructive, good identification of molecular functional groups present within the substance	Expensive, weak spectral signal
Laser-induced Breakdown Spectroscopy	Real-time, simultaneous multi-element analysis, simple detection process, wide detection range	Low resolution, unstable signal strength, baseline drift, substrate effect phenomenon
Nuclear Magnetic Resonance	Simple operation, little damage to the sample, fast detection speeds	Lower resolution, higher environmental requirements
Terahertz Spectroscopy	Strong penetrating power, unique electromagnetic waves can detect the physicochemical information inside the material	Low sensitivity, difficult to detect low-level substances

## Data Availability

No new data were created or analyzed in this study. Data sharing is not applicable to this article.

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
