# Peer review of "Intelligent Rapid Detection Techniques for Low-Content Components in Fruits and Vegetables: A Comprehensive Review"

_foods, 2024, doi:10.3390/foods13071116_

Round 1

Reviewer 1 Report

Comments and Suggestions for Authors

The review entitled "Intelligent Rapid Detection Techniques for Low-Content Components in Fruits and Vegetables: A Comprehensive Review " presents an interesting bibliographic review of the different existing techniques and methods for the detection of minority components using analytical techniques other than those usually used for this purpose.

1.- Introduction: The first two paragraphs are very similar. They repeat information.  Delete one of them.

2.- Line 49: Instead of indicating "substances" put "metals", since the techniques described below are for the determination of metals.

3.- Line 69: What do you mean by the term "solids". It is not clear. Explain.

4.- Lines 95-96: Pesticides or pesticides are not used to control weeds. Herbicides are needed for weed control.

5.- Linea100: In addition to pesticides, include herbicides, fungicides...

6.- Section 3.1: Reduce basic IR operating theory and describe the three forms of measurement indicated, as these may be more unfamiliar to the reader.

7.- Lines 128-129: Clarify. At the beginning it does not indicate "diffuse transmission" and then it does. Clarify and unify terminology.

8.- Line 155: Clarify. To detect fruits and vegetables? It is not understood. I think they meant to indicate: to detect contamination in fruits and vegetables.

9.- Line 160: In HSI cameras can be used in different areas of the electromagnetic spectrum. There are near infrared cameras, Raman cameras…., so it is not opposed to infrared. Clarify this sentence.

10.- Lines 252-255: What do you mean by integrating NMR and IR. The reference cited is impossible to consult for many readers.

11.- Lines 539-544: These concepts have already been mentioned above. Do not repeat them.

12.- Lines 597-602: Again a repeated concept.

13.- Lines 603-604: Thiophan-ate-methyl is not a pesticide. It is a fungicide

14.- Lines 674-677: Eliminate all references whose applications do not refer to fruits and vegetables.

Author Response

Dear reviewer,

Thank you very much for taking time out of your busy schedule to review this manuscript. Responses to each comment are included in the attached document.

Yours sincerely,

Sai Xu

Reviewer 2 Report

Comments and Suggestions for Authors

The manuscript presented for evaluation describes novel detections methods for food analysis.

This manuscript may be a good starting point, but in my opinion it is not suitable for publication in its present form. It should be rewritten, as information on techniques is repeated, e.g. in lines 142-143 and 413-417. Also, the description of techniques and their potential application should be combined. Many sections are unclear and confusing. Methods to be applied to assays require data processing or model preparation, which needs to be explained in the text.

The principles of the techniques presented should be described in more detail. In some cases it is difficult to understand what the method is based on. In some cases, it is necessary to further investigate what it is based on. It is a review article. It should be basic and reliable knowledge to others.

Many abbreviations are not explained in the text.

Some methods described in the manuscript are not well suited for the analysis of food compounds, but rather for texture, structure or other physical properties. These methods also require advanced data analysis, calibration or transformation to be suitable for quantitative or qualitative analysis.

Electronic nose or electronic tongue are not suitable for the detection of compounds present in small quantities, but are rather developed to replace sensory analysis and give the result in terms of odour or taste parameters, not compounds.

Some specific comments are presented below

Line 33 Why you mentioned carotenoids. Other low content compounds may be important. Also, I understand supplementation as an intervention or addition of low content compounds to food or diet.

Lines 38-48 This is a repetition of the previous paragraph.

Line 69 What is the difference between sugars and solids?

Lines 71-73 What about the building function of minerals?

Lines 74-75 What is the main function of vitamins? That substances are needed for normal cell function.

Line 87-88 What does it mean: 'To ensure the efficient use of light energy for energy conversion'. How is iron involved in photosynthesis? I think it is involved in chlorophyll synthesis.

Line 100-102 What about other health effects of pesticides? Pesticides can have direct toxic effects.

Line 102 "Pesticide residues are absorbed by the liver during plant growth" I do not understand this sentence. What plant has a liver?

Lines 112-115 How could the detection of low levels affect dietary habits? I do not understand the context. The aim of the research is not clear to me. Please rewrite these sentences.

Line 117 Give an abbreviation of this technique.

Line 118 - 133 This description sounds strange to me. What is the principle of NIR spectroscopy? How can we detect and recognise the molecular structure of compounds?

As far as I know, bonds formed between some of the atoms can absorb radiation at certain frequencies. By detecting this absorption, it is possible to describe the chemical composition of an unknown mixture or food.

Line 138-139 What does it mean: 'absence of pollution'?

Line 139-140 Is this technique useful for qualitative or quantitative analysis? Fatty acids are compounds with a low content?

Line 142 Add an abbreviation to the title.

Lines 143-148 What is the main idea of the HSI technique? Is it based on spectral images (e.g. NIR images)?

Line 172 - 180 This information should be improved. What exactly is the Raman scattering effect? When does it occur? What is light source and wavelength?

Line 188 Explain abbreviation that appears for the first time in the text.

Line 194 Give abbreviation of this technique.

Line 222 Change the title of the figure to something more appropriate. Maybe Laser-induced breakdown spectroscopy system scheme.

Line 224-237 Only hydrogen protons could be detected? Give more information about the principle of analysis.

Line 238 NMR is used in the energy industry? What about medical purposes?

Line 250 What does 'environmental requirements' mean?

Line 258 Give an abbreviation for this technique.

Line 262 This sentence is a repetition of the previous one.

Lines 265 - 267 THz spectroscopy generally detects very slow vibrations of molecules.

Lines 274-284 In the previous text, the advantages of the technique were only mentioned. Here they are emphasised point by point.

Line 291 Intelligent sensors or intelligent sensors?

Line 293-294 Artificial intelligence need not be involved in intelligent sensor technology.

Line 326 Nutrients are chemical compounds (such as protein, fat, carbohydrate, vitamin or mineral) found in food. These compounds are used by the body to function and grow. These compounds are usually not low in content. And I am not convinced that soluble solids can be included in nutrients.

Line 334 What are CARS, RF or MSC+2-DER? A short explanation and introduction is needed.

Line 334-352 Please emphasise that data processing, theoretical model and calibration are necessary for compound detection by NIR.

Line 335-336 What is a successive projection algorithm and why is it used?

Line 358 What is 'artificially ripened fruit'? Is it related to compound detection?

Line 374 In my opinion N, P and Mg are macro minerals.

Line 386 Detection or quantification? The presence of these compounds is quite obvious.

Lines 426-446 Again model is needed for further analysis. In my opinion the model could not detect anything.

Line 452 - 472 Confusing for me. Please rewrite it.

Line 455 The results showed consistent responses of the different potato varieties to the treatments and light conditions. But what were the responses? HSI image or what?

Line 457 This method shows promise for predicting chlorophyll content in potatoes. Based on what? You have potatoes. How can you predict the chlorophyll content of this plant?

Line 486 What is ANN?

Line 507 Fungus are not compounds.

Line 510-511 Are Aspergillus and Fusarium moulds?

Line 524 Recognition accuracy of what? Mould or aflatoxin?

Line 584 In my opinion these are not trace elements. What does mineral detection have in common with blood sugar reduction?

Line 587 What is ASS?

Author Response

Dear reviewer

Thank you very much for taking time out of your busy schedule to review this manuscript.  Responses to each comment are included in the attached document.

Yours sincerely,

Sai Xu

Round 2

Reviewer 2 Report

Comments and Suggestions for Authors

The authors are making some efforts to improve the manuscript. But in my opinion it still needs improvement.

Line 48 - 50 For me, conventional methods are spectrophotometric or titration. These methods might not be sufficient for the correct detection of low content compounds.

Line 332 - 333 This technique could be suitable for detecting the authenticity of food?

In my opinion, this chapter should be divided according to the groups of compounds that can be detected by these techniques, e.g. health-promoting compounds, pigments, harmful compounds, etc.

Author Response

Dear reviewer

Thank you very much for taking time out of your busy schedule to review this manuscript.  Responses to each comment are included in the attached document.

 Comments 1: Line 48 - 50 For me, conventional methods are spectrophotometric or titration. These methods might not be sufficient for the correct detection of low content compounds.

Response 1: Thank you for your advice. AAS, AFS and ICP-MS are widely used in the field of food component detection because of their high sensitivity and ability to detect low concentrations of ingredients. In the technical application section below, the chemical detection methods used in many research institutes include AAS, AFS, and ICP-MS technologies. In this paragraph, I will add spectrophotometric or titration to the traditional chemical composition analysis method, and change the ‘components’ of the article to ‘low-content components’, so as to fit the title of the article better. The corresponding paragraph in the revised version is 48-49 lines.

Comments 2: Line 332 - 333 This technique could be suitable for detecting the authenticity of food?

Response 2: Thank you for your question. NMR can help identify different ingredients in food and determine their relative content. This is useful for detecting adulteration, which often introduces additional ingredients or changes the proportions of the original ingredients. And if food contains undeclared additives or unexpected contaminants, NMR can help identify and monitor their presence.

Comments 3: In my opinion, this chapter should be divided according to the groups of compounds that can be detected by these techniques, e.g. health-promoting compounds, pigments, harmful compounds, etc.

Response 3: Thank you very much for your advice. We have changed the title content in the corresponding chapter to make the title more relevant to the content of the paper.

Yours sincerely,

Sai Xu